# Current Insights into Antibiotic Resistance in Uropathogenic *Escherichia coli* and Interventions Using Selected Bioactive Phytochemicals

**DOI:** 10.3390/antibiotics14121242

**Published:** 2025-12-08

**Authors:** Bożena Futoma-Kołoch, Jolanta Sarowska, Mohamed Abd El-Salam, David Miñana-Galbis, Barbora Drabová, Katarzyna Guz-Regner, Paula Wiśniewska, Vivien Kryniewska

**Affiliations:** 1Department of Microbiology, Faculty of Biological Sciences, University of Wrocław, Przybyszewskiego 63–77, 51-148 Wroclaw, Poland; 2Department of Medical Biology, Faculty of Nursing and Midwifery, Wrocław Medical University, Chałubińskiego 4, 50-368 Wroclaw, Poland; jolanta.sarowska@umw.edu.pl; 3Department of Pharmacognosy, Faculty of Pharmacy, Delta University for Science and Technology, International Coastal Road, Gamasa 11152, Egypt; 4Research Group in Hypoxia, Dietetics, Nutrition, and Kinanthropometry (GIHDiNuC), Department of Health, TecnoCampus School of Health Sciences, Pompeu Fabra University, 08302 Barcelona, Catalonia, Spain; 5Department of Biology, Health and Environment, Section of Microbiology, Faculty of Pharmacy and Food Sciences, Universitat de Barcelona, 08028 Barcelona, Catalonia, Spain; 6Department of Plant Origin Food Science, University of Veterinary Sciences, Palackeho 1946/1, 612-42 Brno, Czech Republic

**Keywords:** uropathogenic *Escherichia coli*, urinary tract infections, antibiotic resistance, phytochemicals, complementary therapies, efflux pump, biofilm

## Abstract

Uropathogenic *Escherichia coli* (UPEC) is the leading cause of urinary tract infections (UTIs) and a major contributor to the global antimicrobial resistance crisis. The increasing prevalence of multidrug-resistant (MDR) strains, including expanded-spectrum β-lactamases (ESBL) and carbapenemase-producing isolates, severely limits treatment options. This review provides an overview on the key molecular mechanisms of UPEC antibiotic resistance, such as enzymatic inactivation, target-site mutations, efflux pump activity, and biofilm formation. Beyond conventional antibiotics, special emphasis is placed on phytochemical strategies as promising alternatives. Flavonoids, alkaloids, terpenoids, and essential oils exhibit antibacterial, anti-adhesive, and antibiofilm properties. These natural bioactive compounds modulate motility, suppress fimbrial expression, inhibit quorum sensing, and enhance antibiotic efficacy, acting both as standalone agents and as adjuvants. Current in vitro and in vivo studies highlight the potential of plant-derived compounds and biologically based therapies to combat UPEC. However, challenges related to standardization, bioavailability, and clinical validation remain unresolved. Integrating molecular mechanistic insights with advanced phytochemical research may offers a sustainable and effective strategy for mitigating UPEC antibiotic resistance.

## 1. Introduction

Urinary tract infections (UTIs) caused by *Escherichia coli* (*E. coli*) are among the most common infections worldwide, accounting for approximately 80% of all cases. Their high prevalence, along with the frequent occurrence of relapses and complications, necessitates the use of antibacterial agents. These infections constitute a serious global health problem [1,2,3], and epidemiological studies indicate that adult females are 20–30 times more likely to develop a UTI than males [4]. It is estimated that more than 50–60% of women will experience at least one episode of UTI during their lifetime [5]. This striking sex-based difference underscores the need for prevention, diagnosis, and management strategies specifically tailored to women (Figure 1).

The species *E. coli*, belonging to the family *Enterobacteriaceae*, is a component of the gut microbiota of animals and humans [6]. These microorganisms also act as commensalist or mutualist symbionts, involved in the synthesis of certain vitamins and the regulation of digestive processes. A total of 186 somatic (O), 80 capsular (K), and 55 flagellar (H) antigens, as well as more than 160 serological types of *E. coli*, have been identified. This species is one of the most common human pathogens, responsible for infections in multiple body systems, including the gastrointestinal tract, the genitourinary system, and systemic diseases such as sepsis and neonatal meningitis. However, the most frequent extraintestinal form of infection, however, is UTI caused by uropathogenic strains of *E. coli* (UPEC) [2]. Although *E. coli* comprises more than 180 O-serogroups, UPEC isolates are typically associated with a more limited core set of serotypes, most commonly O1, O2, O4, O6, O7, O8, O15, O16, O18, O21, O22, O25, O75 and O83. Of these, O25, O15 and O8 are the most encountered in UTIs. Associations between virulence traits and antibiotic resistance have been observed in specific somatic serogroups of *E. coli*, emphasizing the importance of serotyping in monitoring UTI epidemiology (Figure 2). Furthermore, *E. coli* strains are highly genetically diverse, and multiplex PCR has revealed eight phylogenetic groups: A, B1, B2, C, D, E, F, and clade I [7]. Commensal *E. coli* strains lacking pathogenic properties, which colonize, among others, the mucosal lining of the gastrointestinal tract, most often belong to groups A or B1 [8]. Pathogenic *E. coli* responsible for intestinal infections represent phylogenetic groups A, B1, or D, whereas *E. coli* causing extraintestinal infections belong to groups B2 and D. Group E is related to group D (including O157:H7), while group F is related to the main group B2. Extraintestinal pathogenic *E. coli* (ExPEC) are facultative pathogens that are part of the normal human gut microbiota. The ExPEC group mainly includes UPEC, neonatal meningitis-associated *E. coli* (NMEC), sepsis-associated *E. coli* (SEPEC), and avian pathogenic *E. coli* (APEC) [9,10]. Recent studies suggest that APEC may pose a potential threat to human health, as strains occurring in poultry may serve as a reservoir of virulence genes for UPEC strains pathogenic to humans [11].

UPEC strains possess numerous virulence factors that allow them to colonize mucosal surfaces, evade host defense mechanisms and induce an inflammatory response. The bacteria colonize the urethra and, through an ascending route, may lead to cystitis. A UTI is defined as the presence of local or systemic symptoms, including cystitis and pyelonephritis [12,13,14]. During bladder mucosa colonization, UPEC strains form microcolonies known as intracellular bacterial communities (IBCs), which exhibit biofilm-like properties that enable bacterial persistence in the host. Biofilm formations is one of the most important virulence factors of UPEC strains, particularly in catheter-associated urinary tract infections (CAUTIs) [13,15].

Moreover, epidemiological data indicate a growing burden of multidrug-resistant (MDR) UPEC in UTIs, which is linked to higher costs, longer hospital stays, and more frequent treatment failures [16]. Antibiotics, as defined by the European Food Safety Authority, selectively inhibit or kill microorganisms at low in vivo concentrations. However, the misuse and overuse of antibiotics in medicine, agriculture, and veterinary practice have accelerated the emergence of resistant bacterial strains [17]. According to the European Committee on Antimicrobial Susceptibility Testing, bacterial isolates are classified as *Susceptible* (*S*), *Susceptible*, *Increased Exposure* (*I*), or *Resistant* (*R*) that refers to isolates that are likely to result in therapeutic failure even when exposure to the drug is increased. In fact, MDR bacteria is a major global public health challenge that has been noted, among other international organizations, by the World Health Organization (WHO), that has published an update of the Bacterial Priority Pathogens List, including third-generation cephalosporin- and carbapenem- resistant *Enterobacterales* in the most critical MDR group [18].

In the context of current antibiotic crisis, bacteriophages are being explored as adjuncts to antibiotics for difficult-to-treat UPEC infections. A well-chracterized example is the lytic phage VB-EcoS-Golestan, isolated from wastewater. Genomic analysis showed no lysogeny-associated genes or antibiotic-resistance genes, supporting its therapeutical potential, although clinical validation is still needed [19]. Phage therapy is becoming a promising approach for the treatment and prevention of difficult-to-eradicate infections, despite certain limitations arising from the high specificity of phages toward bacterial species or even strains. Unfortunately, this strain-dependent specificity of phages may result in the selection of phage-resistant UPEC, including MDR *E. coli* strains, and recurrence of UTI [2,20]. Differences in UPEC susceptibility to phages are linked to mutations in DNA sequences involved in LPS biosynthesis and assembly, especially in the core region, which alter bacterial surface properties, increase invasiveness, and enhance biofilm formation. Phage-resistant *E. coli* strains may show greater susceptibility to some conventional antimicrobials targeting the cell membrane and display slower growth under conditions resembling healthy human urine. Other regulatory systems contributing to phage resistance cannot be excluded [21]. Intensive efforts have focused on developing effective phage cocktails with defined biological properties against UTI pathogens. Such preparations, including phage enzymes, offer promising prospects for targeted therapy due to rapid and sustained eradication of bacteria and biofilms and a low probability of resistance [2]. Their application together with antibiotics/chemotherapeutics (β-lactams, carbapenems, aminoglycosides, fluoroquinolones, sulfonamides), even at sub-inhibitory doses, most often produces a synergistic effect. This increases the efficacy against both susceptible and MDR pathogens in multispecies biofilms [2]. In this case, the reduction of bacterial populations and biofilm mass results from the combined action of therapeutic agents causing simultaneous structural and functional disruptions of pathogen cellular components (biological membranes, cell wall, fimbriae, adhesins) and cellular processes related to bacterial growth and development, along with enhanced biological activity of lytic phages (better penetration, facilitated adsorption, and host cell entry). This manifests as increased membrane permeabilization via enzymatic action, degradation or inhibition of extracellular polymeric matrix (EPS) biofilm matrix production, changes in strain virulence, and improved efficacy of antibiotics/chemotherapeutics [2,22] (Figure 3).

Bacterial elimination continues to be significantly hindered due to their ability to form biofilms. Biofilms, as organized structures of bacterial cells, produce an EPS and are capable of colonizing both abiotic and biotic surfaces [23,24]. Bacteria embedded in the extracellular matrix, composed of a mixture of polymeric compounds such as proteins, polysaccharides, lipids, and nucleic acids, exist in proximity and form channels for the transport of nutrients, oxygen, water, and enzymes. This provides favorable conditions unavailable to free-living bacteria in planktonic cultures. The most important factor influencing the reduced susceptibility of bacterial biofilms to bioactive substances is the exopolysaccharide matrix, which, in addition to these functions, enables horizontal gene transfer and cell-to-cell communication [25].

The transition of bacterial cells from planktonic growth to biofilm growth is associated with altered expression of numerous genes encoding both virulence factors and regulatory proteins. The ability of *E. coli* to form biofilms is influenced by expression of the *flu* gene, which is linked to *E. coli* cell aggregation [11]. Numerous studies have demonstrated that disruption of purine biosynthesis in *E. coli* reduces intracellular levels of cyclic di-GMP (c-di-GMP), an important regulator of many cellular processes, including motility, biofilm formation, and virulence. Reduced c-di-GMP levels in *E. coli* result in decreased biofilm production and curli fiber synthesis, which also highlights the involvement of multiple other regulatory factors activated via c-di-GMP. At each stage of biofilm formation, certain genes play key roles, such as fimbrial genes *fimB* and *fimE*, and *csgE* genes encoding proteins involved in curli fiber transport outside the bacterial cell [7]. Nevertheless, other studies have shown the significant role of many additional genes, including *luxS*, *rpoS*, *sdiA*, and *rcsA*, in biofilm development and antibiotic resistance acquisition [26].

Additionally, the biofilm matrix protects embedded cells from antimicrobial agents, environmental stress, and host immunity [26,27]. In this state, bacteria residing in biofilms are up to 1000 times more resistant to drugs or antimicrobials compared to planktonic forms. The main causes of this tolerance are morphological and physiological changes in bacteria inhabiting biofilms. These modifications lead to reduced metabolic activity, slower growth rates, and increased resistance to stress responses, triggered by mechanisms such as limited antimicrobial penetration, entry into a persister state, efflux pump activation, and horizontal gene transfer (HGT) [13,26].

The multilayer structure of biofilms and the high density of microorganisms mean that bacterial cells may display different physiological properties depending on their position within the biofilm [11,28]. Cells located in deeper layers of the biofilm are exposed to conditions of lower oxygen concentration, accumulation of metabolites, and nutrient deprivation. These cells enter a phase of slowed growth and metabolic dormancy, which is associated with decreased susceptibility to antimicrobial agents. In contrast, bacteria colonizing the surface layers of biofilms experience optimal growth conditions and exhibit a phenotype like planktonic cells, along with greater antibiotic susceptibility [27].

Bacteria in the deeper layers of biofilms show increased production of various hydrolytic enzymes and extracellular polysaccharides, as well as enhanced biosynthesis of efflux pumps. The function of efflux pumps is to actively expel antimicrobial substances, including active components of disinfectants and antibiotics, thereby reducing their intracellular concentration and consequently limiting their bactericidal activity [26]. Furthermore, due to the presence of the polysaccharide matrix, the diffusion of antibiotics into the inner layers of the biofilm is impeded, preventing these compounds from reaching all cells at effective concentrations. It is believed that within the deeper layers of biofilms, resistant strains to antimicrobial agents may be selectively enriched [28].

One of the key challenges of contemporary microbiology and pharmacology is the growing resistance of bacteria to antibiotics and biocides, as well as the search for new antibacterial substances that would be both effective and safe. Outside Europe, antibiotics are often used as feed additives for poultry at subtherapeutic levels to stimulate growth and combat gastrointestinal infections. The discovered link between the use of subthreshold concentrations of broad-spectrum tetracyclines in agriculture and the increase in human bacterial isolates with acquired resistance led to the introduction of a ban on the use of tetracyclines as growth-promoting additives in poultry farming in Europe in the early 1970s [29]. Through mutations, resistance gene transfer, biofilm formation, cell wall modifications, or efflux pump overexpression [30,31,32,33,34,35], bacteria are becoming increasingly difficult to eradicate. Consequently, the emergence of hard-to-treat bacterial infections is becoming a growing challenge in human medicine [36]. Due to cases of biofilm-forming strains resistant to disinfectants and antibiotics, it is crucial to develop alternative elimination strategies.

In the search for alternatives to conventional antibiotics for UTI management, particularly in cases involving MDR pathogens, natural products and plant-derived compounds have gained significant attention as promising sources of antibiofilm and anti-infective agents. While several emerging strategies such as phage-derived enzymes, depolymerases, and endolysins are being explored for their ability to disrupt biofilms, these approaches remain largely experimental, strain-dependent, and require extensive evaluation of safety, pharmacodynamics, and delivery before routine clinical implementation [2,37,38,39,40]. Similarly, other enzymatic tools, including proteases, peptidases, lyases, amylases, lipases, and cellulases, have been proposed to degrade biofilm matrix components and enhance biofilm susceptibility, mainly in industrial settings, but their translational potential for clinical UTI treatment is still limited [24,25,41].

Given these constraints, medicinal plants and their phytochemicals represent a more accessible and biologically versatile alternative, with broad antimicrobial, antibiofilm, and antivirulence activities. Numerous plant extracts contain bioactive groups such as phenols, terpenoids, alkaloids, lectins, polypeptides, polyacetylenes, and essential oils that can inhibit biofilm formation, including multispecies biofilms through multiple mechanisms. These include disrupting pathogen cell membranes, reducing substrate availability, and interfering with adhesins, cell wall components, and surface proteins of both bacterial and fungal pathogens [24,41]. Increasing evidence also shows that phytochemicals can function as natural QS inhibitors. Compounds such as ajoene and allicin from *Allium sativum* L. as well as galloylquinic acids from *Copaifer lucens* L. suppress QS systems (AHL and autoinducer-2 mediated pathways) and have been shown to inhibit biofilm formation by *Pseudomonas* spp. and *Vibrio* spp. [41]. This multifunctional activity highlights plant-derived metabolites as highly promising candidates for combating biofilms and antibiotic resistance in UTIs.

It has also been discovered that phytochemicals can act as natural QS inhibitors by interfering with autoinducer-1 (AHL) and autoinducer-2 signaling. Garlic extracts effectively inhibit biofilm formation by *Pseudomonas* spp. and *Vibrio* spp. by disrupting QS communication systems [41]. The compounds ajoene and allicin from *Allium sativum* L. effectively inhibit rhamnolipid production, bacterial adhesion in early biofilm development, and interfere with QS-mediated bacterial communication, thus reducing virulence. Another example is carvacrol (a monoterpenoid) from *Origanum vulgare* L., which inhibits post-translational *lasI* processes, limiting AHL production, the main activator of QS. Meanwhile, epigallocatechin-3-gallate (EGCG) from *Camellia sinensis* L. inhibits the expression of *csgA*, *csgB*, and *csgD* genes encoding proteins involved in curli fiber production and assembly in *E. coli*, while enhancing degradation of sigma factor RpoS, an important regulator of adaptive stress responses. According to many researchers, combining such antibiofilm phytochemicals with antibiotics may effectively combat pathogens in humans and animals [42].

There is also growing interest in secondary metabolites with anti-adhesive and antibiofilm properties produced by algae and cyanobacteria. Extracts from *Chlamydomonas* species, for example, reduced surface hydrophobicity of Gram-negative *Pseudomonas aeruginosa* rods, thereby hindering adhesion to biotic and abiotic surfaces, and inhibited EPS secretion and biofilm formation. Certain algal compounds acting as acyl-homoserine lactone (AHL) analogs influenced QS-regulated gene expression, reducing biofilm and virulence of *P. aeruginosa*. Likewise, polyphenols from *Sargassum muticum* showed antibiofilm activity by inhibiting biofilm formation by Gram-negative *E. coli* and *P. aeruginosa*, while *Oscillatoria* spp. extract enhanced with silver nanoparticles exhibited strong antibacterial effects against pathogens such as *Staphylococcus aureus*, *E. coli*, *P. aeruginosa*, *Salmonella* Typhi, and *Bacillus cereus*, along with strong antibiofilm activity. Such multicomponent preparations may represent potential antimicrobial agents with applications in medicine, agriculture, and industry [24].

As noted in study by Al-Maddboly et al. (2025) *P. aeruginosa* within biofilms exhibits reduced metabolic activity and altered gene expression, which enhances its tolerance to antibiotics and promotes survival during treatment [43]. A key mechanism underlying both tolerance and persistence is dormancy, a physiological state in which bacterial cells reduce or halt metabolic activity. Because many antibiotics target active cellular processes, dormant cells are less susceptible. In tolerance, dormancy may be induced across the entire population, while in persistence, it is limited to a small subpopulation. Dormancy is associated with decreased energy production (ATP), and suppression of replication, transcription, and translation-contributing to the overall survival of the bacteria under antimicrobial pressure [44]. Understanding the distinctions between resistance, tolerance, and persistence is crucial for developing new therapeutic strategies, improving diagnostic accuracy, and mitigating the rise of treatment-refractory infections.

## 2. *E. coli* Strategies for Antibiotic Resistance

Bacterial cells exposed to biocides can develop numerous adaptations that result in tolerance or resistance to a wide range of toxic compounds. Reduced bacterial susceptibility to biocides may arise from intrinsic resistance mechanisms or from acquired resistance mechanisms involving horizontal gene transfer (HGT) (e.g., phages, plasmids, transposons), or it may be generated because of mutations [45]. Intrinsic resistance refers to an innate feature of a given group of microorganisms or a trait developed through adaptive changes [46]. Bacterial cells that do not exhibit natural resistance can acquire this trait by developing mechanisms that decrease their sensitivity to biocides. These include changes in the ratio of saturated to unsaturated fatty acids, destabilization and loss of LPS from the cell wall of Gram-negative bacteria, repression of porin biosynthesis, efflux pump overexpression, as well as alteration of target sites and enzymatic inactivation of the biocide [13,47].

The WHO has recognized antimicrobial resistance as one of the greatest contemporary threats to public health, which has contributed to rapid advances in research on bacterial resistance and monitoring of this phenomenon, especially in the case of zoonotic bacteria, both pathogenic and commensal, that serve as reservoirs of resistance genes [40,48,49]. MDR arises when bacteria accumulate multiple resistance genes or mutations, reducing susceptibility to several antibiotic classes. Alarmingly, extensively drug-resistant (XDR) and even pandrug-resistant (PDR) strains are increasingly being detected, with PDR phenotypes emerging through the combined acquisition of diverse resistance mechanisms [26,50,51]. Sustained exposure of *E. coli* to sub-inhibitory biocide levels can select for mutations that promote antibiotic cross-resistance, while mobile-element-encoded, substrate-specific efflux mechanisms further increase this risk by carrying additional resistance genes that facilitate the spread of multidrug resistance [45,52]. Increased use of antibiotics for therapeutic purposes, as well as for prophylaxis in agriculture and animal husbandry, has contributed to the spread of resistant bacteria, with animals serving as reservoirs of such strains and transmitting them to humans either directly or indirectly through the food chain [53,54].

Gram-negative bacteria commonly employ four mechanisms of drug resistance and, at the same time, exhibit a high potential for resistance gene transfer through HGT. These mechanisms include: (i) reduced drug uptake, (ii) modification of the drug target, (iii) drug inactivation, and (iv) active drug efflux [55,56]. Resistance to one antimicrobial agent can, in some cases, extend to related drugs through mechanisms such as target modification, efflux pump overexpression or reduced membrane permeability. These forms of cross-resistance may broaden resistance profiles and decrease susceptibility to multiple classes of antimicrobial compounds (chemotherapeutics or disinfectants). In the latter case, this indicates the involvement of shared mechanisms and strategies in resistance expression linked to bacterial stress responses [45].

Resistance mechanisms acting against β-lactams (penicillins, cephalosporins, monobactams, carbapenems) include active efflux and the production of β-lactamases, which inactivate the drugs. Due to the mechanism of action of aminoglycoside antibiotics (amikacin, gentamicin, tobramycin), resistance is based on aminoglycoside-modifying enzymes, modification of the 16S rRNA target, and active efflux. Reduced uptake and active efflux lower the effectiveness of tetracyclines, chloramphenicol, and fluoroquinolones (ciprofloxacin, norfloxacin), whose activity may also be impaired by modified targets—DNA gyrase or topoisomerase IV. Modification of the target enzyme underlies resistance to metabolic pathway inhibitors such as trimethoprim/sulfamethoxazole [57].

UPEC strains employ several mechanisms of antimicrobial resistance, most commonly through the acquisition of mobile resistance genes (e.g., ESBLs, carbapenemases, aminoglycoside-modifying enzymes). In contrast, chromosomal mutations affecting global regulators of efflux and permeability, such as *marR*, *acrR*, *envZ*, *ompR* and *nlpD*, occur less frequently in clinical UPEC isolates but can contribute to broader resistance profiles when present. These mutations may lead to cross-resistance to antibiotics and biocides by altering key physiological processes, including increased expression of efflux pumps, reduced outer-membrane permeability and enhanced cell envelope stability. For example, mutations in *marR* derepress the *marRAB* operon, allowing MarA to activate the *acrAB* efflux system through repression of the local regulator AcrR. Similarly, alterations in EnvZ can result in constitutive phosphorylation of OmpR, shifting the OmpF/OmpC porin ratio and limiting entry of various antimicrobial compounds. Mutations in *nlpD*, particularly loss of the dLytM domain, increase tolerance to cell-wall stressors by enhancing resistance to lysis. Although these regulatory mutations can lead to decreased susceptibility to multiple antibiotic classes-including β-lactams, fluoroquinolones, aminoglycosides, sulfonamides, tetracyclines and chloramphenicol-they should be considered accessory rather than predominant resistance mechanisms in UPEC. Importantly, such mutations may also influence virulence traits, for example by reducing the expression of type 1 fimbriae [45].

Prolonged exposure to increasing doses of antibiotics results in increased resistance, while simultaneous exposure to different types of drugs leads to the selection of MD strains. Subinhibitory levels of various antibiotics, including β-lactams, can induce cell wall stress and stimulate biofilm formation. Amoxicillin is one of several agents reported to produce such effects [58]. Other studies have also shown that sub-inhibitory concentrations of antibiotics (ciprofloxacin, ampicillin, and gentamicin) modulate the virulence of UPEC strains [59,60].

In Gram-negative bacteria, antimicrobial resistance genes located on plasmids, transposons, or chromosomal DNA often assemble into integrons, which contain genes encoding β-lactamases (*bla*) as well as determinants of resistance to aminoglycosides, sulfonamides, tetracyclines, or quinolones. Integron gene cassettes identified in *E. coli* include dihydrofolate reductase genes *dfrA1* and *dfrA7*, conferring resistance to trimethoprim, as well as *bla*, *tet*(*B*), and *qnrB* genes, conferring resistance to β-lactam antibiotics, tetracyclines, and quinolone antibiotics, respectively [61].

Empirical therapy for UTIs should be guided by the distinction between uncomplicated and complicated infections, local resistance patterns, and current clinical guidelines. For uncomplicated lower UTIs, first-line agents include nitrofurantoin and fosfomycin, while trimethoprim/sulfamethoxazole (co-trimoxazole) may be used only in regions where resistance levels remain acceptably low. In more severe or complicated infections, or when oral therapy is unsuitable, aminoglycosides (e.g., gentamicin) and β-lactams-including amidinopenicillin derivatives (pivmecillinam) and third-generation cephalosporins (e.g., ceftazidime, ceftriaxone)-as well as fluoroquinolones may serve as empirical options, provided that their use is supported by local epidemiological data [13,45,62,63]. Nevertheless, resistance to many of these agents has increased markedly over the past decades, reducing the reliability of empirical regimens.

A major contributor to this trend is the global expansion of the UPEC ST131 lineage, currently recognized as the most epidemiologically successful extraintestinal pathogenic *E. coli* clone worldwide. ST131 combines a characteristic virulence profile, including enhanced adhesiveness associated with specific *fimH* alleles, with the ability to acquire and disseminate MDR determinants, most notably CTX-M-15, but also TEM-1, TEM-24, and SHV-12 [13,64,65]. The convergence of high virulence and extensive antimicrobial resistance in this lineage poses a substantial challenge to empirical UTI management and represents a significant global public health threat.

In a Polish single-centre study from January 2013 to December 2015 [42], 498 hospitalized patients with UTIs were examined: *E. coli* accounted for 72.9% of isolates and 8.0% of these were ESBL-producers. A Danish outpatient study by Córdoba et al. (2017) conducted in 2014–2015 found ESBL-producing *E. coli* in 6.0% of 505 cases [66]. In contrast, a 2018 Iranian cohort of 126 *E. coli* isolates from hospitalized UTI patients reported a MDR rate of 77.8% and ESBL-production of 54.8 % [67]. The dominant resistance pattern included ampicillin, trimethoprim/sulfamethoxazole, nalidixic acid, ceftazidime, and ciprofloxacin, and the proportion of ESBL-producing strains reached as high as 54.8%. This was likely since the Iranian study group consisted of nephrology ward patients, with ESBL producers being predominantly hospital-associated strains. These studies provide useful historical benchmarks; however, more recent surveillance indicates a substantially higher global burden. For instance, a 2025 meta-analysis estimated a pooled prevalence of ESBL-producing *E. coli* in clinical isolates at 42.1 % (95% CI: 37.3–46.9) across 56,324 isolates from 2015–2024 [68].

In a study by Katongole et al. (2020) [69] conducted in Uganda, biofilm-forming capacity and antimicrobial susceptibility patterns of MDR *E. coli* UPEC isolates were evaluated. According to the results, 62.5% (125/200) of strains were biofilm producers, while 78% (156/200) showed MDR. The isolates exhibited the highest resistance to trimethoprim, sulfamethoxazole, and amoxicillin (93%), followed by gentamicin (87%). A meta-analysis in a systematic literature review indicated frequent occurrence of biofilm-producing *E. coli* UPEC strains in UTIs, with more than 82% of studies reporting a direct association between biofilm development and antibiotic resistance. Multiple reports have shown an increased biofilm-producing capacity among UPEC strains carrying MDR and ESBL resistance mechanisms [26].

Carbapenem-resistant *E. coli* (CR-*E. coli*) represent one of the most clinically challenging pathogen groups, as carbapenemases effectively hydrolyze all β-lactam antibiotics, including last-resort agents used in severe infections [34,70]. Their emergence reflects advanced bacterial resistance strategies involving acquisition of mobile carbapenemase genes (e.g., *blaOXA-48*, *blaKPC*, *blaVIM*, *blaNDM*) combined with additional mechanisms such as porin loss or efflux pump upregulation, which together markedly reduce intracellular antibiotic concentrations [26,71]. These CR strains must be distinguished from MDR *E. coli*, which exhibit resistance to at least three antimicrobial classes but do not necessarily harbor carbapenemases. In some settings, MDR lineages belonging to highly virulent phylogroups such as B2 have also been identified, although this association is not typical globally and likely reflects local clonal expansion rather than a generalizable trend [72,73]. Importantly, CR-*E. coli* often co-carry determinants conferring resistance to fluoroquinolones, aminoglycosides or third-generation cephalosporins, enabling them to withstand multiple therapeutic options and facilitating persistent colonization of the urinary tract. From a mechanistic standpoint, the combination of enzyme-mediated β-lactam hydrolysis, altered permeability and efflux-based drug extrusion represents a highly effective bacterial strategy for surviving antimicrobial pressure and contributes to the rapid spread of these high-risk clones [74,75].

## 3. Virulence and Adaptive Mechanisms of UPEC

A close association has been observed between the coexistence of virulence factors and MDR in UPEC, particularly in strains carrying F plasmids, which frequently act as vehicles for both resistance and virulence genes [76,77]. In addition to classical genetic resistance, UPEC may withstand antibiotic exposure through phenotypic strategies such as tolerance and persistence. Tolerance reflects a transient ability of the bacterial population to survive antibiotic stress without an increase in MIC, whereas persistence involves a small subpopulation of dormant cells capable of surviving high drug concentrations and subsequently repopulating once treatment ends. These non-heritable adaptations, often linked to metabolic dormancy and reduced ATP generation, contribute to treatment failure and recurrent UTIs [44].

Virulence factors also play a significant role in UPEC survival and pathogenicity. Toxins encoded by *cnf1*, *hlyA*, *sat* and *vat*—including hemolysins and the cytotoxic necrotizing factor CNF1—induce cytotoxicity, promote tissue invasion and damage erythrocytes, leukocytes and renal epithelial cells, triggering strong inflammatory responses. Additionally, many MDR UPEC strains exhibit enhanced siderophore activity, including production of enterobactin, aerobactin, yersiniabactin and salmochelin, which, together with hemolysins, facilitate successful colonization of the urinary tract and survival despite host immune defenses. Invasins such as SisA and SisB further support early infection by suppressing host immune responses [3,13,77].

Accurate characterization of UPEC strains, including their resistance profiles and virulence-associated traits, is essential for effective clinical management. A clear understanding of the mechanisms underlying MDR and pathogenicity can support optimized therapeutic decision-making and guide the development of new treatment strategies for UTIs.

### 3.1. Basis of Two-Component Systems in Bacterial Stress and Drug Resistance

Bacterial survival depends on the ability to sense and respond to a wide range of environmental stimuli. This process is primarily mediated by regulatory proteins, among which two-component signal transduction systems (TCSs) play a central role. TCSs enable rapid and adaptive physiological responses, controlling processes such as cell envelope remodeling, virulence, metabolism, biofilm formation and antimicrobial resistance, thereby serving as a major strategy by which bacteria detect and adjust to environmental changes [78]. As highlighted by Jung et al. (2018) [79], these systems are highly modular and versatile, enabling bacteria to process diverse physicochemical signals-from osmolarity and pH to nutrient availability and antimicrobial stress. Through this architecture, TCSs enable rapid and adaptive physiological responses, controlling processes such as cell envelope remodeling, virulence, metabolism, biofilm formation and antimicrobial resistance, and thus represent a major mechanism by which bacteria adjust to fluctuating environmental conditions. The TCS comprises two protein components: (i) the sensor histidine kinase (SHK), an integral membrane protein that functions as a sensor and transduces signals across the cytoplasmic membrane, and (ii) the response regulator (RR), which receives the phosphate group transferred from the SHK’s phosphorylated histidine to a conserved aspartate residue within the RR. Phosphorylated RR typically acts as a transcription factor, modulating the expression of target genes. Signal recognition occurs via the sensor domain of the SHK, which triggers activation of the kinase domain.

The kinase domain of a sensor histidine kinase (SHK) is composed of two conserved subdomains: (i) the DHp (dimerization and histidine phosphotransfer) domain, which harbors the invariant histidine residue serving as the phosphorylation site, and (ii) the CA (catalytic and ATP-binding) domain, which binds ATP and catalyzes the autophosphorylation reaction. Upon signal perception by the periplasmic or cytoplasmic sensor domain, conformational changes propagate through the transmembrane helices to the kinase core, triggering ATP-dependent autophosphorylation on the DHp histidine. The resulting high-energy phospho-histidine intermediate subsequently donates its phosphoryl group to the conserved aspartate within the receiver (REC) domain of the response regulator (RR). Phosphorylation typically stabilizes the active conformation of the RR, enhancing its DNA-binding affinity and enabling precise transcriptional activation or repression of target genes [80].

As emphasized by Bhagirath et al. (2019) [81], numerous TCSs directly contribute to antibiotic resistance in Gram-negative pathogens by regulating efflux pump expression, modifying the outer membrane, altering LPS structure, and activating stress-response pathways that mitigate antibiotic toxicity. Examples such as PmrA/PmrB, PhoP/PhoQ or BaeSR illustrate how TCS-mediated transcriptional reprogramming can reduce membrane permeability, enhance drug efflux or modify target structures, collectively promoting survival under antimicrobial pressure. Investigations by Choi et al. (2024) identified mutations in *marR*, *acrR* and the TCS gene *envZ*, which were associated with MDR connected to ampicillin, ciprofloxacin, trimethoprim, chloramphenicol and tetracycline in UPEC strains [45]. TCSs allow bacteria to sense environmental signals and rapidly adjust gene expression through a conserved phosphorelay mechanism. These regulatory pathways may significantly contribute to antimicrobial resistance, including MDR in UPEC.

### 3.2. Role of Adhesins in UPEC Infections

UPEC expresses a broad repertoire of adherence factors that enhance bacterial attachment, colonization and persistence within the urinary tract. These adhesins function not only as mediators of surface binding but also as virulence determinants that trigger host–pathogen signaling pathways, support secretion of bacterial effectors and promote epithelial invasion. Consistent with their pathogenic lifestyle, UPEC strains harbor significantly more fimbrial gene clusters than commensal *E. coli* isolates. Through these diverse adhesins, UPEC recognize bladder and kidney epithelial cells, immune cells, erythrocytes and extracellular matrix components. The specific anatomical site of infection is largely determined by the adhesin repertoire expressed during intimate contact with host tissues [82,83].

Among virulence factors, adhesins encoded by *afa*, *csh*, *fimH*, *fimP*, *kpsMTII*, *pap*, *sfa* and *traT* are most frequently highlighted. These fimbrial and non-fimbrial structures are crucial for attachment, colonization and intracellular invasion, facilitating the establishment of persistent bacterial communities within the urinary tract [77,84]. Within this large adhesin arsenal, type 1 fimbriae and P fimbriae are the most common and best characterized. Type 1 fimbriae are expressed by approximately 80–100% of UPEC isolates and are highly abundant on the cell surface, with an estimated 100–500 pili per cell. Their clinical relevance is underscored by studies from Mobley et al. (1987) [85], showing significantly higher expression among strains causing persistent rather than transient infections. P fimbriae represent another major adherence system, strongly associated with both bladder and kidney infections through specific recognition of P blood group antigens on uroepithelial cells. Notably, expression of type 1 and P fimbriae is coordinately but inversely regulated, allowing UPEC to adjust attachment properties to different microenvironments within the urinary tract [85,86,87].

The Dr adhesin family also plays a significant role in UPEC pathogenicity and comprises both fimbrial (Dr) and nonfimbrial adhesins (AFA-I, AFA-II, AFA-IV, Nfa-I and Dr-II). These structures mediate strong binding to epithelial surfaces and support bacterial invasion, contributing to persistent and complicated UTIs. In CAUTIs, both specific and nonspecific adhesins likely contribute to early colonization of catheter surfaces. However, the precise fimbrial structures responsible for catheter adherence have not yet been fully identified [77,88,89,90,91]. Genomic analyses indicate that more than 60% of UPEC strains carry 5–15 chaperone–usher (CU) fimbrial operons. Among them, type 1, type 3, type 9, S, P, F1C and Auf fimbriae constitute the most frequent CU fibers in UPEC pathotypes [92,93,94]. Studies identify type 1, P, S, F1C, F9, Iha, and Dr adhesins as key determinants of UPEC colonization within the urinary tract. Moreover, cell-surface hydrophobicity is present in approximately 80% of isolates, with alongside frequent detection of fimbrial and afimbrial adhesins such as fimH (75.3%) and fimP (35.6%) [77,95].

## 4. Efflux-Mediated Antibiotic Resistance and the Search for Natural Inhibitors

One of the mechanisms of antibiotic resistance in bacteria is the active extrusion of substances from the cell by efflux pumps. These are protein complexes located in the cytoplasmic membrane of both Gram-positive and Gram-negative bacteria. Efflux pumps expel toxic substances from the cell using energy [30,96]. They emerged during evolution to allow bacteria to survive in their natural ecological niches, protecting them, for example, from toxins produced by other bacteria [97]. Examples of substances expelled by efflux pumps include disinfectants, reactive oxygen species, toxic by-products of biochemical reactions, detergents, dyes, antibiotics, and metal ions. Individual pumps may be substrate-specific or responsible for the removal of antibiotics belonging to multiple classes [98].

Efflux pumps are classified into six major families. Their classification is based on amino acid sequence similarity, secondary structure, and phylogenetic relationships [99]. Transporters can also be categorized by the type of energy used to expel antimicrobial agents outside the cell. Proteins of the ABC family use energy derived from ATP hydrolysis, while the other groups rely on the proton motive force (arising from the proton gradient) or on a sodium ion gradient [98]. The major facilitator superfamily (MF) is one of the most widespread, transporting a variety of substrates across the membrane. The small multidrug resistance (SMR) family primarily handles low-molecular-weight compounds, while the multi-antimicrobial extrusion (MATE) family confers resistance against diverse antimicrobial agents. The resistance-nodulation-cell division (RND) superfamily is particularly significant in Gram-negative bacteria due to its broad substrate specificity and contribution to MDR. ATP-binding cassette (ABC) transporters utilize the energy from ATP hydrolysis to actively pump out toxic compounds. Finally, the proteobacterial antimicrobial compound efflux (PACE) family represents a more recently identified group, specifically involved in exporting antimicrobial compounds in proteobacteria [100]. Genes encoding efflux pump proteins may be located on the bacterial chromosome, making this an intrinsic trait, or on plasmids, where they can be acquired through genetic material transfer during conjugation, transduction, or transformation. The best-studied efflux system is AcrAB-TolC, belonging to the RND family, present in Gram-negative bacteria, including *E. coli*. AcrB, the largest component of the system, recognizes substrates and facilitates their transport into the periplasmic space, functioning as the transport protein located in the inner cytoplasmic membrane. TolC enables substrate expulsion outside the cell, while AcrA connects these two components. The AcrAB-TolC protein complex is found in enteric bacteria of the *Enterobacteriaceae* family, allowing them to survive in environments with high bile salt concentrations [7,30]. Overexpression of the *AcrA* gene has been linked to resistance to the β-lactam antibiotic ertapenem. Efflux pumps are also associated with resistance to certain fluoroquinolones, trimethoprim, and nitrofurantoin [101].

Efflux pump activation is environment-dependent, as they are not constitutively expressed by bacteria. Their expression is induced by harmful substances and stress conditions. One method of studying efflux pump activity is the use of their inhibitors, followed by measuring the concentration of the substance expelled from the cell by a given protein [102]. For a compound to be classified as an efflux pump inhibitor, it must lack intrinsic antibacterial activity and act specifically on a given efflux pump [101]. The first efflux pump inhibitor discovered was MC-207,110, whose activity against RND pumps in *P. aeruginosa* and *E. coli* was identified in 2001 [102]. Current research is focused on inhibitors derived from plants, particularly alkaloids, flavonoids, and polyphenols [103].

An example of an alkaloid is reserpine, derived from *Rauwolfia serpentina*. This inhibitor binds to pumps belonging to the MF and RND families. When combined with norfloxacin, it exhibited four times stronger activity. However, reserpine has never been used on a larger scale due to its toxic effects on kidney cells. Other plant-derived compounds with efflux pump inhibitory properties include piperine (from *Piper nigrum*) and harmaline (from *Peganum harmala*). Both compounds increased ciprofloxacin accumulation in *S. aureus* by inhibiting the NorA efflux pump [103].

Studies on efflux pump inhibitors have also been conducted in UPEC strains. In a study published last year, the PAβN inhibitor was used to assess efflux pump activity in UPEC strains. A significant, two-fold reduction in MIC values for ciprofloxacin was observed with this inhibitor. In three UPEC strains, the MIC decreased four-fold. The same publication analyzed efflux pump gene expression levels: each strain showed overexpression of at least one efflux pump gene, further confirming the link between antibiotic resistance and efflux pumps. The most frequently observed were AcrA and AcrB, with overexpression recorded in 11 and 15 out of 16 strains, respectively. Overexpression of all tested efflux pump genes—*AcrA*, *AcrB*, *MdfA*, and *NorE*—was observed in 11 out of 16 strains [104].

In another study, the efflux pump inhibitor chlorpromazine was used to test the cause of antibiotic resistance by measuring bacterial sensitivity before and after adding the compound. Efflux pump–mediated resistance was present in 92% of the UPEC strains studied. Among MDR *E. coli* strains, efflux-dependent resistance was as high as 98.7%. In 95% of these strains, efflux pumps expelled more than one type of antibiotic, showing the broad action of this mechanism—all such strains were MDR. Genes forming the AcrAB-TolC system were also detected in these UPEC strains. The genes *acrA*, *acrB*, and *tolC* were found in 74% of UPEC and MDR isolates. In the remaining strains, variant systems were observed, with absence of one of the two genes (*acrA* or *acrB*) in 8% and 11% of UPEC strains and 1% of MDR isolates, respectively. Cases of absence of two or three AcrAB-TolC genes were recorded in fewer than 6% of *E. coli* strains, with the lack of two genes being more frequently associated with increased sensitivity to tetracycline [105]. Research on the action of efflux pump inhibitors in bacterial cells is currently one of the strategies to combat antibiotic resistance.

## 5. Phytochemicals-Based Approaches to Overcome Antibiotic Resistance in UPEC

One of the main causes of recurrent and chronic urinary tract infections caused by UPEC is biofilm production. In the past, a correlation between biofilm formation and increased antibiotic resistance has been reported; bacteria embedded in biofilms exhibit higher resistance to antibiotics compared to planktonic cells. The first step in biofilm formation is the synthesis of curli protein fibers, which promote adhesion of bacterial cells to solid surfaces. In a study of 175 UPEC isolates, 46% showed curli fiber production, of which 99% also displayed an antibiotic resistance phenotype. Biofilm production was also evaluated, with high production activity observed in 44% of UPEC strains and 48% of MDR strains. These results confirm a strong relationship between MDR-type resistance and biofilm production [106].

Some researchers use the term *recalcitrance* to encompass both tolerance and persistence, as these transient, non-heritable phenotypic adaptations enable bacterial survival during antibiotic treatment. Recalcitrant or dormant cells can withstand exposure to multiple antibiotic classes, contributing to recurrent and treatment-refractory infections. Dormancy, characterized by reduced ATP production and suppression of replication and transcription, decreases antibiotic susceptibility because most antimicrobials target active cellular processes. In tolerant populations, dormancy may occur across the entire bacterial community, whereas in persistence it is confined to a small subpopulation of cells. Understanding these distinctions is crucial for developing effective strategies to eradicate biofilm-associated and chronic infections caused by UPEC [43,44].

Figure 4 illustrates the principal mechanisms by which selected phytochemicals counteract UPEC resistance and virulence, including efflux pump inhibition, anti-adhesive activity, suppression of motility and biofilm formation, and membrane disruption. Complementary to this schematic overview, Table 1 summarizes representative phytochemicals, their primary targets, mechanisms of action, and observed outcomes, providing detailed experimental evidence for their antimicrobial and antivirulence effects against UPEC.

### 5.1. Anti-Adhesive Phytochemicals

Biofilm formation is a major virulence and survival strategy of UPEC and is strongly associated with chronic and recurrent UTIs. In the study by Fattahi et al. [106], 92% of *E. coli* isolates from hospitalized UTI patients produced biofilm, underlining how common this phenotype is in clinical strains. Biofilm production protects bacteria from environmental stressors—including antibiotics—and is therefore a key driver of treatment failure and infection recurrence [107]. Biofilm-associated cells exhibit markedly increased tolerance to antimicrobial agents, with MIC and minimal bactericidal concentration (MBC) values reported to be 10–1000-fold higher than those of planktonic cells [108]. This enhanced tolerance results from multiple factors: the protective exopolysaccharide matrix, the presence of metabolically dormant subpopulations, induction of resistance mechanisms, production of antibiotic-degrading enzymes and efficient horizontal gene transfer within the biofilm [13,103,108,109]. Mutations in regulatory genes controlling major efflux pump families (MFS, SMR, MATE, ABC, RND, DMT) and reduced outer membrane permeability further broaden resistance to multiple antimicrobial classes [13,110,111,112]. Persister cells, characterized by low metabolic activity and high tolerance to antibiotics, can survive treatment and reseed the biofilm, contributing to recurrent infections [13,113,114].

Given the limited efficacy of conventional antibiotics against biofilm-associated UPEC, novel eradication strategies are being explored. Approaches such as phage–antibiotic synergy, photodynamic therapy and ultrasound-assisted disruption have demonstrated promising antibiofilm effects but remain largely experimental [76,115,116]. In parallel, increasing attention has turned toward naturally derived compounds as adjuncts or alternatives to standard therapy. Many plant-derived compounds are classified as GRAS and contain diverse bioactive molecules—terpenes, polyphenols and phytosterols—that can exert antibacterial, anti-adhesive and antibiofilm effects through multiple mechanisms, including interference with adhesion structures, modulation of motility, disruption of membranes, inhibition of curli and EPS synthesis or enhancement of antibiotic penetration [25,117,118].

Several plant extracts have shown activity against UPEC in vitro. Bean seed extract (PPX) reduced adhesion of UPEC strain NU14 to bladder epithelial cells, primarily by altering host cell susceptibility rather than directly inhibiting bacterial growth [119]. Fruit extracts, particularly from quince, also significantly decreased UPEC adhesion in a concentration-dependent manner [120]. Wojnicz et al. demonstrated that extracts from *Equisetum arvense*, *Hydrocotyle glabra*, *Galium odoratum*, *Betula pendula*, *Vaccinium vitis-idaea* and *Urtica dioica* differ in their anti-adhesive potential against UPEC, with some preparations additionally inhibiting curli fiber synthesis [121,122]. These findings support the concept that selected phytochemicals can attenuate key virulence traits—such as adhesion and biofilm formation—rather than acting purely as classical bactericidal agents.

Overall, the growing interest in plant-derived compounds reflects the urgent need for complementary strategies in UTI management. By targeting adhesion, motility and biofilm formation—central determinants of UPEC persistence and antibiotic tolerance—phytochemicals and other natural products may help improve treatment outcomes and reduce reliance on conventional antibiotics [117,118].

### 5.2. Phytochemicals as Inhibitors of Motility and Biofilm Formation

Swimming and swarming motility are considered important UPEC virulence traits and contribute to the initial stages of host colonization. In addition, *E. coli* produces curli fibers-extracellular amyloid structures also found in *Salmonella*, *Citrobacter* and *Enterobacter*-which play a crucial role in adhesion and biofilm formation. Curli assembly is followed by production of extracellular polymeric substances (EPS), supporting biofilm maturation. Motility enables bacteria to navigate liquid environments or swarm across solid surfaces; however, these two forms of movement are differentially regulated. Swimming is performed by individual cells, whereas swarming represents a collective behavior regulated by quorum sensing [24,41,94,123].

Motility is especially important during the early stages of biofilm establishment, aiding colonization of new surfaces such as uroepithelial cells or catheter materials. Once UPEC become established, flagellar gene expression is typically downregulated, favoring a transition from motile to sessile forms better suited for persistent biofilm-associated infections. Flagellum-mediated motility may also facilitate bacterial ascent from catheter surfaces to the bladder and upper urinary tract. Several regulatory proteins modulate motility in *E. coli*: YcgR slows flagellar rotation by interacting with the MotA/FliG/FliM complex, while CsrA, OmpR and RcsB influence transcription of the master regulator *flhDC*. A detailed analysis of motility regulation in bacteria is provided in the review by Guttenplan and Kearns [123].

Given the role of motility in UPEC pathogenesis and biofilm development, numerous studies have examined plant-derived compounds for their ability to modulate these processes. Several phytochemicals show promising antimicrobial and anti-biofilm activity against UPEC [87,123,124]. In vitro assays using UPEC strain CFT073 demonstrated that cranberry extracts inhibited swimming motility by downregulating *fliC*, confirmed by qRT-PCR and electron microscopy showing reduced flagella formation [125]. Similarly, Dusane et al. (2014) showed that sub-inhibitory concentrations of the alkaloids piperine (*Piper nigrum*) and reserpine (*Rauwolfia serpentina*) decreased expression of *fliC*, *motA* and *motB*, reduced swimming and swarming motilities, and increased expression of adhesin genes (*fimA*, *papA*, *uvrY*) [117]. Combined treatment of phytochemicals with ciprofloxacin or azithromycin enhanced eradication of pre-formed UPEC biofilms, likely due to improved antibiotic penetration.

Other studies also demonstrate anti-motility and anti-adherence properties of plant extracts. Extracts from *Betula pendula* and *Urtica dioica* inhibited motility of clinical *E. coli* isolates from pyelonephritis patients. Promising effects have also been observed for *H. glabra*, *V. vitis-idaea*, and *J. quince* extracts, although antibacterial activity varied [103]. Cranberry and barberry are among the most widely investigated sources of phytochemicals with activity against UPEC biofilms and adhesion; however, it has also been shown that antibiotics (ciprofloxacin, amikacin and colistin) at sub-inhibitory concentrations can reduce UPEC biofilm production [124,126]. Ranfaing et al. (2018) [127] found that cranberry combined with propolis strongly inhibited motility and biofilm formation in UPEC strains causing cystitis, pyelonephritis and asymptomatic bacteriuria, although cranberry alone was less effective against some isolates. De Llano et al. (2015) [128] confirmed anti-adhesive properties of cranberry phenolic metabolites, which limited UPEC colonization and UTI development. Specific cranberry flavonoids—particularly B-ring-substituted flavones and flavonols—interfere with FimH-mediated adhesion to bladder epithelial cells [129].

Additional phytochemicals display varied anti-UPEC effects. Cinnamaldehyde inhibits biofilm formation by downregulating adhesion and invasion genes [130]. *p*-Coumaric acid disrupts cell membrane synthesis and exhibits antibacterial activity even against fluoroquinolone-resistant clinical isolates [131,132]. In contrast, ferulic acid showed no reduction in UPEC biofilm biomass [133]. Polysaccharides from *Vaccaria segetalis* demonstrated anti-adhesive and anti-invasive effects against UPEC CFT073 in vitro and in vivo [134]. Lewis et al. (2024) tested four phenolic compounds—CAPE, resveratrol, catechin and EGCG—and found that resveratrol inhibited adhesion and, together with CAPE, reduced UPEC invasion [135].

Essential oils (EOs) form another category of phytochemicals with growing interest. Their chemical composition varies by species, geographic origin, developmental stage and extraction method [136]. Many EOs display broad-spectrum antimicrobial properties and may reduce reliance on synthetic antibiotics [137]. Mangalagiri et al. (2021) evaluated seven EOs and reported that lemongrass oil exhibited bactericidal activity comparable to ciprofloxacin and gentamicin, while peppermint oil was ineffective; notably, no resistance emerged after prolonged exposure to sub-MIC concentrations [138]. However, resistance induction by EOs remains debated. McMahon et al. (2007) found that repeated exposure of *E. coli* to sub-MIC tea tree oil decreased susceptibility to multiple antibiotics, with MIC values increasing up to four-fold [139].

In the context of rising antibiotic resistance, plant-derived compounds represent a promising adjunct or alternative strategy for UTI treatment. However, their use requires careful evaluation of chemical composition, antimicrobial activity, mechanisms of action and the potential for resistance development.

**Table 1 antibiotics-14-01242-t001:** Examples of plant-derived phytochemicals with anti-adhesive, anti-motility, and anti-biofilm activities against UPEC. The listed compounds interfere with bacterial virulence factors through enhancing antibiotic efficacy or preventing biofilm-associated resistance.

Phytochemical	Primary Target	Mechanism of Action	Observed Outcome	References
Cranberry (*Vaccinium macrocarpon*)	Adhesion, Motility, Biofilm	Inhibition of FimH-mediated adhesion; downregulation of *fliC*; reduction of flagellar motility	Decrease in UPEC adhesion and colonization; decrease in biofilm formation	[125,128,129]
Propolis + Cranberry extract	Motility, Biofilm	Synergistic inhibition of flagellar motility and EPS-associated biofilm formation	Strong impact on the motility and the biofilm formation of UPEC	[127]
Cinnamaldehyde (CNMA)	Adhesion, Biofilm	Inhibits UPEC biofilm formation by suppressing motility, reducing fimbriae, and damaging the bacterial membrane	Strong reduction of UPEC biofilm, fimbriae, motility and growth	[130]
p-Coumaric acid	Growth, Membrane integrity	Disruption of bacterial cell membranes and possible binding to bacterial genomic DNA	Loss of membrane integrity, disruption of cellular functions, inhibition of growth	[131,132]
Allyl isothiocyanate (AITC) 2-Phenylethyl isothiocyanate (PEITC)	Growth	Disruption of the bacterial cell membrane	Potassium leakage, altered surface properties, growth inhibition, bactericidal activity	[133]
Resveratrol	Adhesion, Invasion	Inhibition of FAK (Y576) phosphorylation and block actin-dependent invasion	Strong inhibition of UPEC invasion with minimal effect on adhesion	[135]
CAPE (caffeic acid phenethyl ester)	Invasion	Inhibition of FAK (Y576) phosphorylation and block actin-dependent invasion	Strong inhibition of UPEC invasion with minimal effect on adhesion	[135]
Catechin, Epigallocatechin gallate—EGCG	Adhesion	More subtle, but still discernable, effects on pFAK (Y576)	Have minimal effects on UPEC adhesion and modest reduction of invasion	[135]
Bean pods extract (*Phaseoli pericarpium*)	Adhesion	Antiadhesive activity against	↓ *E. coli* NU14 adhesion to T24 human bladder cells (dose-dependent)	[119]
Quince (*Cydonia oblonga*) extract	Adhesion	—	↓ Bacterial adhesion; moderate antibacterial activity	[120]
*Betula pendula*, *Urtica dioica* extracts	Motility	—	↓ UPEC motility in *E. coli* clinical isolates	[121]
*Vaccaria segetalis* polysaccharides	Adhesion, Invasion, Motility	Inhibits UPEC adhesion, invasion and motility by downregulating fimbrial adhesins, TLR signalling and uroplakin expression	Significant reduction of UPEC adhesion, invasion, motility, and bladder colonization	[134]
Lemongrass essential oil (*Cymbopogon flexuosus*)	Growth	Possible cell membrane disruption	Antibacterial activity effect against *E. coli*; no resistance after 30 passages	[138]
Tea tree oil (*Melaleuca* *alternifolia*)	Growth/ Resistance modulation	Sub-MIC exposure may induce cross-resistance to antibiotics	Potential risk of reduced antibiotic susceptibility in *E. coli* after repeated exposure	[139]
Piperine, Reserpine	Motility, Biofilm	Downregulation of *motA*, *motB*; upregulation of *fimA*, *papA*, *uvrY*	↓ Motility; ↑ adhesion; enhanced antibiotic efficacy	[117]

↓ and ↑ arrows denote decreased and increased, respectively.

## 6. Summary

UPEC remains the leading cause of UTIs worldwide and a major driver of MDR. A deeper understanding of UPEC pathogenesis will shed light on recurrent infections and the emergence of antimicrobial resistance, facilitating the identification of novel targets for effective therapeutic development. The alarming increase in ESBL and carbapenemase-producing strains underscores the urgent need for new therapeutic strategies. The main resistance mechanisms in UPEC include enzymatic inactivation of antibiotics, target-site mutations, efflux pump overexpression, and biofilm formation, often modulated by complex regulatory systems such as two-component signaling pathways that link resistance and virulence. Phytochemicals and natural products represent promising adjunctive approaches to antibiotics. Flavonoids, terpenoids, alkaloids, phenolic acids, and essential oils act through multiple mechanisms. These categories of bioactive plant secondary metabolites inhibit bacterial adhesion and motility, disrupt quorum sensing, suppress fimbrial expression, impair biofilm maturation, and potentiate antibiotic efficacy. Compounds such as EGCG, carvacrol, and ajoene exemplify how natural products can modulate resistance pathways and virulence determinants. Recent findings also highlight the potential of algae- and cyanobacteria-derived metabolites with anti-adhesive and anti-quorum sensing effects, broadening the scope of natural antibiofilm strategies. Despite these advances, translating phytochemical activity from in vitro to in vivo and clinical settings remains challenging. Critical issues include variability in extract composition, lack of standardization, limited bioavailability, potential toxicity, and insufficient mechanistic and pharmacokinetic data. Future research should focus on standardizing extract preparation, identifying molecular targets through omics-based tools, and validating efficacy in relevant infection models and clinical trials. In addition, several phytochemicals show synergy with conventional antibiotics, supporting their potential use as part of combined therapeutic strategies. Integrating molecular microbiology, phytochemistry, and pharmacology will be essential to develop safe, effective, and sustainable therapies for MDR UPEC infections. Such multidisciplinary efforts will not only enhance our understanding of bacterial resistance and virulence but also pave the way toward clinically relevant phytotherapeutic interventions and improved management of recurrent and chronic UTIs.

## Figures and Tables

**Figure 1 antibiotics-14-01242-f001:**
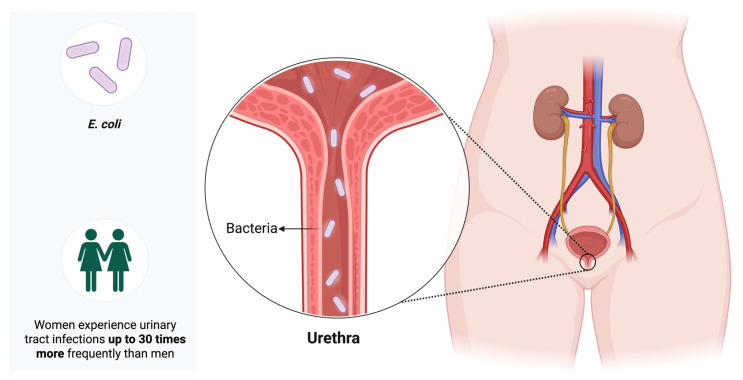
**Uropathogenic *Escherichia coli* (UPEC) is the principal causative agent of urinary tract infections (UTIs).** It ascends from the periurethral or intestinal microbiota through the urethra to colonize the urinary tract. UTIs are much more frequent in women, who are affected up to 30 times more often than men.

**Figure 2 antibiotics-14-01242-f002:**
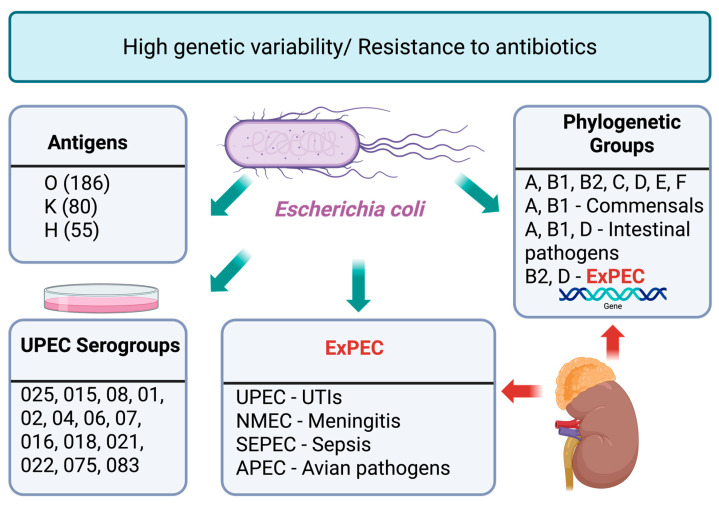
Genetic diversity, phylogenetic classification, and pathogenic potential of *Escherichia coli* in urinary tract infections, and clinical relevance. *E. coli* exhibits high genetic variability and widespread antibiotic resistance. Based on antigenic composition, strains are differentiated by O, K, and H antigens and classified into various serogroups. Phylogenetically, *E. coli* strains are divided into groups A, B1, B2, C, D, E, and F, with groups A and B1 typically associated with commensal or intestinal pathogens, while B2 and D correspond to extraintestinal pathogenic *E. coli* (ExPEC). ExPEC pathotypes include uropathogenic *E. coli* (UPEC) causing urinary tract infections (UTIs), neonatal meningitis *E. coli* (NMEC), sepsis-associated *E. coli* (SEPEC), and avian pathogenic *E. coli* (APEC).

**Figure 3 antibiotics-14-01242-f003:**
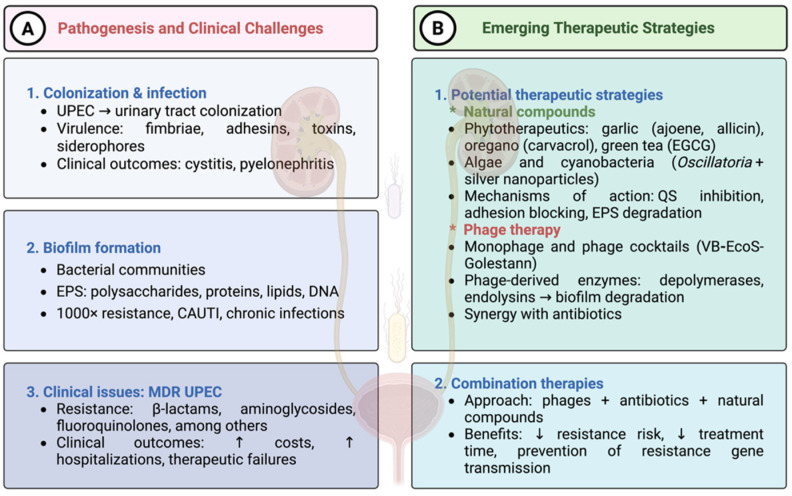
**Pathogenesis, clinical challenges, and emerging therapeutic strategies against uropathogenic *Escherichia coli* (UPEC).** (**1A**) UPEC initiates urinary tract colonization through fimbriae, adhesins, toxins, and siderophores, causing infections such as cystitis and pyelonephritis. (**2A**) Biofilm formation enhances bacterial persistence and can increase resistance up to 1000-fold, contributing to catheter-associated and chronic infections. (**3A**) Multidrug-resistant (MDR) UPEC strains exhibit resistance to β-lactams, aminoglycosides, and fluoroquinolones, leading to higher treatment costs, prolonged hospitalizations, and frequent therapeutic failures. (**B**) Novel therapeutic approaches include: (**1B**) Natural compounds, such as phytotherapeutics, algal extracts, and nanoparticles, which inhibit quorum sensing, block adhesion, and promote biofilm disruption. (**2B**) Phage therapy using either monophasic or cocktail formulations, as well as phage-derived enzymes that degrade biofilms. Combination strategies integrating phages, antibiotics, and natural agents may help reduce resistance development, shorten treatment duration, and prevent the spread of resistance genes. Abbreviations: UPEC—uropathogenic *Escherichia coli*; CAUTI—catheter-associated urinary tract infection; EPS—extracellular polymeric substances; QS—quorum sensing; MDR—multidrug-resistant; EGCG—epigallocatechin gallate.

**Figure 4 antibiotics-14-01242-f004:**
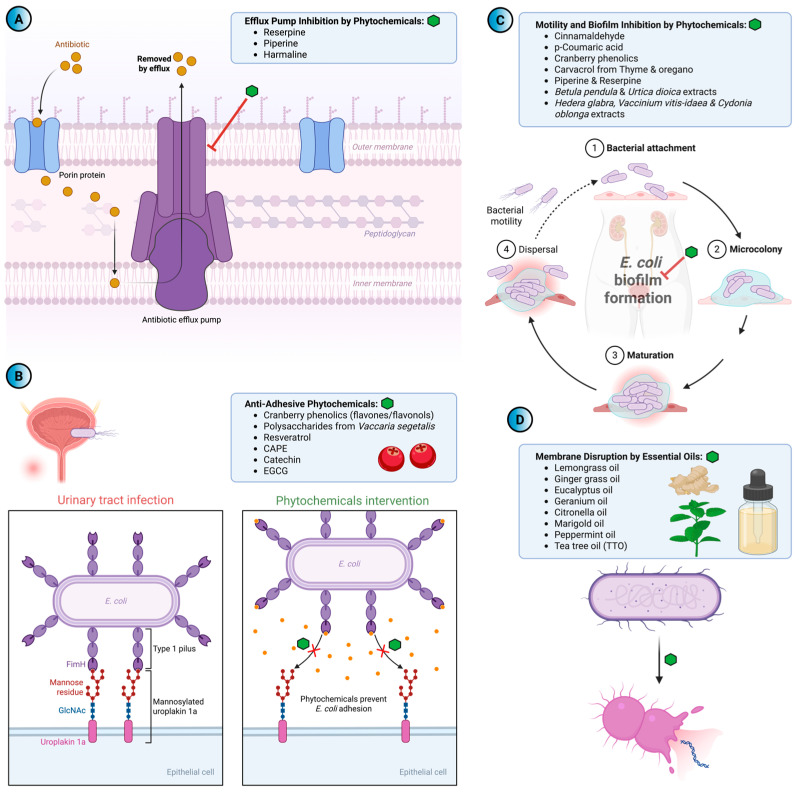
Phytochemical compounds targeting uropathogenic *E. coli* (UPEC) through multiple mechanisms. (**A**) Efflux pump inhibition: Selected phytochemicals and synthetic compounds such as reserpine, piperine, and harmaline act by inhibiting major efflux pump families (MF, RND), enhancing intracellular antibiotic accumulation and restoring susceptibility. (**B**) Anti-adhesive phytochemicals: Cranberry phenolics, resveratrol, and other plant-derived compounds interfere with UPEC adhesion and invasion, primarily through inhibition of FimH-mediated binding to uroepithelial cells. (**C**) Motility and biofilm inhibition: natural compounds including cinnamaldehyde, p-coumaric acid, carvacrol, and extracts from *Betula pendula*, *Urtica dioica*, *Hedera helix* var. glabra, *Vaccinium vitis*-idaea, and *Cydonia oblonga* suppress flagellar motility and biofilm formation via transcriptional modulation of motility and adhesion genes. (**D**) Membrane disruptors and essential oils: volatile oils from *Cymbopogon citratus*, *Pelargonium* spp., *Eucalyptus* spp., *Calendula* spp., and *Melaleuca alternifolia* exhibit bactericidal effects by disrupting bacterial membranes, with MBCs comparable to conventional antibiotics.

## Data Availability

No new data were created or analyzed in this study.

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
