# Peer review of "Current Insights into Antibiotic Resistance in Uropathogenic *Escherichia coli* and Interventions Using Selected Bioactive Phytochemicals"

_antibiotics, 2025, doi:10.3390/antibiotics14121242_

Round 1
Reviewer 1 Report
Comments and Suggestions for Authors
The manuscript presents an extensive and detailed review of antibiotic resistance mechanisms in uropathogenic Escherichia coli (UPEC) and the promising use of plant-derived compounds as alternative or adjunct therapeutic strategies. The topic is of significant scientific and clinical relevance given the growing burden of multidrug-resistant (MDR) UTI pathogens. The manuscript is well-structured and demonstrates substantial literature coverage, including both classical and recent findings.
However, the current version requires major revisions before it can be accepted for publication. The main issues relate to (a) figure quality and legibility, (b) excessive text density with limited synthesis or critical analysis, (c) inconsistent formatting of references, and (d) occasional grammatical and stylistic errors that affect clarity and readability.
- The strengths of the manuscript include comprehensive coverage of UPEC virulence factors and resistance mechanisms, inclusion of translationally relevant discussions on phytochemicals and essential oils with anti-adhesive, anti-motility, and antibiofilm activity, and presentation of in vitro and in vivo findings where available.
- One major issue is the quality of figures and visuals. Several figures are low-resolution and text within them is difficult to read. High-quality, vector-based images are recommended. Including a schematic summarizing UPEC resistance mechanisms and another showing plant-derived compounds’ actions would improve clarity. A summary table listing phytochemicals, their targets (adhesion, motility, biofilm), and observed outcomes would strengthen the manuscript.
- Language and grammar require attention. Several long and complex sentences reduce readability. Professional English language editing is recommended. Minor typographical errors (e.g., “ini-tiate,” “Hydrocotyle,” repeated words) should be corrected. Terminology should be standardized throughout (e.g., “plant-derived compounds,” “phytochemicals,” “phytocompounds”).
- The organization and flow of the manuscript could be improved. Some sections, particularly 4.2–6, read as descriptive lists rather than synthesized analysis. Summarizing key findings and highlighting trends would enhance readability. Short transition sentences between sections would improve logical flow.
- Critical analysis is limited in some areas. The manuscript should discuss limitations and gaps in current research, such as differences between in vitro and in vivo efficacy of phytochemicals, challenges in clinical translation (bioavailability, toxicity, standardization), and mechanistic insights into TCS pathways and efflux regulation.
- Referencing and formatting need attention. References should be consistently formatted and sequential. All abbreviations (e.g., EPS, ECM, MBEC) should be defined at first mention. Concentration units should be standardized (mg/mL vs. mg·mL⁻¹).
- The manuscript would benefit from greater scientific depth. Analytical comparisons between classes of phytochemicals (e.g., flavonoids vs. terpenes) should be highlighted, along with connections between bacterial motility, biofilm formation, and efflux-mediated resistance.
- The conclusion and discussion of future directions should be expanded to include forward-looking recommendations for research priorities and potential clinical translation. The integration of natural compounds with conventional therapy could be emphasized.
- Minor comments include correcting grammatical issues and typos throughout the text, replacing outdated or inconsistent terminology for adhesins and fimbriae, ensuring consistent subheadings, numbering, and formatting, and adding cross-references to figures or tables where appropriate.
- Overall, the manuscript has strong potential to make a valuable contribution to the field. However, it requires major revision to improve clarity, synthesis, figure quality, critical analysis, and discussion of future directions before it is suitable for publication.
Comments on the Quality of English Language
-
Several long and complex sentences reduce readability. Professional English language editing is recommended.
-
Minor typographical errors (e.g., “initiate,” “Hydrocotyle,” repeated words) should be corrected.
-
Standardize terminology (e.g., “plant-derived compounds,” “phytochemicals,” “phytocompounds”). Correct small grammatical issues and typos throughout the text.
Author Response
We thank the Reviewer 1 for these insightful comments. We agree that the points raised were well founded, and we carefully considered each one. Addressing them has significantly improved the clarity, consistency, and overall quality of the manuscript.
General Reviewer’s comments:
However, the current version requires major revisions before it can be accepted for publication. The main issues relate to (a) figure quality and legibility,
We thank the Reviewer for this comment. All figures have been replaced with high-resolution, vector-based versions. Additionally, we have added two new schematic illustrations: Figure 1. Overview of urinary tract infection (UTI) pathogenesis and epidemiology, Figure 4. Phytochemical compounds targeting uropathogenic E. coli (UPEC) through multiple mechanisms, and (iii) Table 1. Examples of plant-derived phytochemicals with anti-adhesive, anti-motility, and anti-biofilm activities against UPEC.
(b) excessive text density with limited synthesis or critical analysis
This suggestion has been fully addressed. The manuscript has been revised to reduce excessive text, improve readability, and include clearer syntheses of key concepts.
(c) inconsistent formatting of references,
We have thoroughly revised the reference list to ensure consistency and compliance with the journal’s formatting guidelines. All numbering and citation styles have been standardized.
and (d) occasional grammatical and stylistic errors that affect clarity and readability
The entire manuscript has undergone detailed language editing. Long and complex sentences were simplified, stylistic issues were corrected, and overall clarity and readability have been improved.
Specific comments:
Comment 2. One major issue is the quality of figures and visuals. Several figures are low-resolution and text within them is difficult to read. High-quality, vector-based images are recommended. Including a schematic summarizing UPEC resistance mechanisms and another showing plant-derived compounds’ actions would improve clarity. A summary table listing phytochemicals, their targets (adhesion, motility, biofilm), and observed outcomes would strengthen the manuscript.
As addressed in the responses above, this concern has been fully incorporated. All figures have been replaced with high-quality vector-based versions, two new schematic illustrations have been added (Figure 1 and Figure 4), and a summary table outlining phytochemicals, their targets, and observed outcomes has been included in the revised manuscript.
Comment 3. Language and grammar require attention. Several long and complex sentences reduce readability. Professional English language editing is recommended. Minor typographical errors (e.g., “ini-tiate,” “Hydrocotyle,” repeated words) should be corrected. Terminology should be standardized throughout (e.g., “plant-derived compounds,” “phytochemicals,” “phytocompounds”).
The entire manuscript has undergone detailed language editing. Long and complex sentences were simplified, stylistic issues were corrected, and overall clarity and readability have been improved. Terminology has been fully standardized throughout the manuscript. We now consistently use “plant-derived compounds” and “phytochemicals,” and outdated or redundant terms related to adhesins and fimbriae have been corrected.
Comment 4. The organization and flow of the manuscript could be improved. Some sections, particularly 4.2–6, read as descriptive lists rather than synthesized analysis. Summarizing key findings and highlighting trends would enhance readability. Short transition sentences between sections would improve logical flow. In addition, we expanded several figure captions to provide clearer explanations of the depicted mechanisms and thereby enhance the reader’s understanding.
We appreciate this helpful suggestion. Sections 4 and 5 have been reorganized to improve logical flow and coherence. Transitional sentences have been added to strengthen the narrative continuity. Moreover Section 2 “Current definitions of resistance, tolerance, and the persistence phenotype of bacteria to antimicrobial agents” has been removed and its content integrated into other relevant chapters to ensure a more streamlined structure – pages 4 and 8.
Comment 5. Critical analysis is limited in some areas. The manuscript should discuss limitations and gaps in current research, such as differences between in vitro and in vivo efficacy of phytochemicals, challenges in clinical translation (bioavailability, toxicity, standardization), and mechanistic insights into TCS pathways and efflux regulation.
We appreciate this comment and agree that distinguishing between in vitro and in vivo findings is an important aspect of evaluating phytochemical activity. While a comprehensive analysis of these differences was not included because it was beyond the primary scope of this review, we do partially address this issue in the Summary section, where we highlight the challenges associated with translating in vitro results to in vivo and clinical settings.
Comment 6. Referencing and formatting need attention. References should be consistently formatted and sequential. All abbreviations (e.g., EPS, ECM, MBEC) should be defined at first mention. Concentration units should be standardized (mg/mL vs. mg·mL⁻¹).
All abbreviations are now defined at first mention. Concentration units were standardized across all sections.
Comment 7. The manuscript would benefit from greater scientific depth. Analytical comparisons between classes of phytochemicals (e.g., flavonoids vs. terpenes) should be highlighted, along with connections between bacterial motility, biofilm formation, and efflux-mediated resistance.
We appreciate this valuable comment regarding the need for stronger analytical comparisons between different classes of phytochemicals. While a comprehensive, side-by-side comparison of phytochemical classes (e.g., flavonoids vs. terpenes) was not incorporated, as this level of classification-focused analysis was beyond the primary scope of the review. Furthermore, the expanded summary table now clearly organizes representative phytochemicals, their primary targets, and observed outcomes, which helps the reader identify functional similarities and differences between compound types in a concise manner. Thus, although not presented as a direct inter-class comparison, the revised manuscript provides clearer cross-cutting insights into how various phytochemical groups modulate adhesion, motility, biofilm formation, and efflux-related resistance. We thank the Reviewer for this suggestion, which helped us improve the interpretative clarity of the manuscript.
Comment 8. The conclusion and discussion of future directions should be expanded to include forward-looking recommendations for research priorities and potential clinical translation. The integration of natural compounds with conventional therapy could be emphasized.
We appreciate this valuable recommendation. The conclusion section “Summary” has been extensively revised to provide clearer forward-looking perspectives. In the updated version, we expanded the discussion of key research priorities, including challenges in translational research, the need for standardization, issues of bioavailability and toxicity, and the importance of validating phytochemical activity in relevant infection models and future clinical studies. In addition, we highlight the potential synergy between phytochemicals and conventional antibiotics.
Comment 9. Minor comments include correcting grammatical issues and typos throughout the text, replacing outdated or inconsistent terminology for adhesins and fimbriae, ensuring consistent subheadings, numbering, and formatting, and adding cross-references to figures or tables where appropriate.
We thank the Reviewer for these helpful observations. All minor grammatical issues and typographical errors have been corrected throughout the manuscript. Outdated or inconsistent terminology related to adhesins, and fimbriae has been replaced with standardized nomenclature. Subheadings, numbering, and formatting have been fully harmonized, and cross-references to figures and tables have been added where appropriate to improve clarity and consistency. In addition, two subsections “3.2. Role of adhesins in UPEC infections” and “5.1. Anti-adhesive phytochemicals” have been reorganized to ensure that their content does not overlap and that each addresses a distinct aspect of the topic.
Reviewer 2 Report
Comments and Suggestions for Authors
Please refer to the PDF file to find the suggestions and comments.
Major comments
- The review is generally comprehensive but requires a more systematic and coherent structure. The current flow of discussion is difficult to follow. At times, similar points are repeated under different subheadings; at other times, discussions of mechanisms and treatments are intermixed.
- I recommend reorganizing the content in a more logical sequence, such as (but not limited to) the following structure:
— Introduction to UPEC
— Definitions of resistance and relevant epidemiology
— Overview of key resistance mechanisms to be discussed (e.g., efflux pumps)
— Virulence and adaptive mechanisms, including their association with resistance (e.g., TCS, adhesins, motility, biofilms)
— Therapeutic strategies organized by targeted mechanisms (e.g., phage therapy, phytochemicals in this section)
- Since the article emphasizes phytochemical strategies, a summary table outlining the mentioned approaches and their associated mechanisms would enhance clarity and readability.
- For example: Section 3—"UPEC: bacterial strategies against biocides and antibiotics”
the content jumps from “intrinsic vs. acquired resistance to biocides” to “efflux pumps,” then “permeability and One Health,” and finally to “resistance mechanisms of GNB and various antibiotics”. The arrangement lacks coherence.
Specific Comments:
The following are some, but not all, of the suggestions/errors found in the article.
- Comment 1
Only 14 serotypes are listed— please clarify or expand.
- Comment 2
Please define the abbreviations in Fig. 1.
What does “ZUM” represent?
- Comment 3
Please provide reference(s).
- Comment 4
“the only possible intervention” is too arbitrary. There are other non-pharmaceutical interventions, such as antimicrobial stewardship program and patient education.
- Comment 5
(a) Correct to VB_ecoS-Golestan, not VB-ecoS-Golestan.
(b) Please provide the reference for the statement and clarify why this particular phage therapy is highlighted over others.
- Comment 6
Please spell out EPS upon first use.
- Comment 7
“as well as the resulting research for…….”: remove “resulting”
- Comment 8
The description regarding phage therapy gives the misleading impression that is is well-established with minimal concerns— please revise for accuracy.
- Comment 9
Please define the abbreviations in Fig. 2.
Typo: VB_ecoS-Golestan
- Comment 10
Please spell out EMBO upon first use.
- Comment 11
Typo
- Comment 12
(a) Clarify: Enterobacterales order or Enterobacteriaceae family?
(b) Provide a more structured of efflux pump classifications and their relationship to MDR/XDR/PDR — consider merging this into the efflux section.
(c) This sentence does not fit the context.
(d) This statement is overly simplified.
- Comment 13
If the section addresses chromosomal mutations affecting efflux pumps and porins that contribute to cross-resistance, clarify that these are not common resistance mechanisms in UPEC. Please introduce the resistance mechanisms more systemically.
- Comment 14
Amoxicillin is neither the only antibiotics promoting biofilm formation nor specifically highlighted in the cited reference.
- Comment 15
The empirical treatment section appears to list antibiotics in a somewhat random manner.
If ST131 is to be mentioned, it should be properly contextualized—highlighting its predominant role in the global dissemination of E. coli, as well as its hallmark traits of co-expressing key virulence determinants and harboring multidrug-resistant ESBL genes.
- Comment 16
The epidemiological reports cited are neither up to date nor reflective of the current clinical landscape.
- Comment 17
MDR and CR E. coli are typically discussed as distinct entities in epidemiological literature. Why is B2 phylogroup mentioned?
— what is the specific relevance of these discussions to the subheading “bacterial strategies” ?
- Comment 18
This paragraph lacks relevance to resistance — consider merging it into the virulence section.
- Comment 19
Please clarify—do you mean: The response regulator (RR), which receives a phosphate group transferred from the SHK’s histidine residue to an aspartate residue “on the RR”?
- Comment 20
For this section designated as “The role of two-component regulatory systems in UPEC antibiotic resistance”—this is one of the few relevant descriptions that correlates to “TCS and resistance”.
Please construct a more relevant and organized discussion.
- Comment 21
Please provide the reference.
- Comment 22
The writing in this section needs refinement to improve readability.
- Comment 23
Consider moving the paragraph to the plant-based approaches section.
- Comment 24
Describe the most common and well-described at the beginning of the section.
- Comment 25
With which plant extracts?
- Comment 26
This section is largely repetitive, please revise.
- Comment 27
This also repeats earlier content.

Author Response
We sincerely thank the Reviewer for the thorough and constructive assessment of our manuscript. The comments were extremely helpful. The revisions substantially improved the clarity and scientific accuracy of the review. Below we provide detailed responses to all points raised.
Major comments
The review is generally comprehensive but requires a more systematic and coherent structure. The current flow of discussion is difficult to follow. At times, similar points are repeated under different subheadings; at other times, discussions of mechanisms and treatments are intermixed.
We fully agree with the Reviewer’s assessment. The manuscript has been extensively reorganized to ensure a clear, systematic, and coherent flow. Overlapping content was removed, and related discussions were consolidated.
I recommend reorganizing the content in a more logical sequence, such as (but not limited to) the following structure:
— Introduction to UPEC
— Definitions of resistance and relevant epidemiology
— Overview of key resistance mechanisms to be discussed (e.g., efflux pumps)
— Virulence and adaptive mechanisms, including their association with resistance (e.g., TCS, adhesins, motility, biofilms)
— Therapeutic strategies organized by targeted mechanisms (e.g., phage therapy, phytochemicals in this section)
Thank you for this helpful suggestion. We restructured the manuscript to keep a balance, according to the suggestions of all Reviewers. To improve coherence, Sections 4 and 5 have been reorganized to improve logical flow and coherence. Transitional sentences have been added to strengthen the narrative continuity. Moreover Section 2 “Current definitions of resistance, tolerance, and the persistence phenotype of bacteria to antimicrobial agents” has been removed and its content integrated into other relevant chapters to ensure a more streamlined structure.
Since the article emphasizes phytochemical strategies, a summary table outlining the mentioned approaches and their associated mechanisms would enhance clarity and readability.
As recommended, we added Table 1, which summarizes the key phytochemical classes, their mechanisms of action, plant sources, and supporting experimental evidence. This significantly improves clarity for readers.
For example: Section 3—"UPEC: bacterial strategies against biocides and antibiotics”
the content jumps from “intrinsic vs. acquired resistance to biocides” to “efflux pumps,” then “permeability and One Health,” and finally to “resistance mechanisms of GNB and various antibiotics”. The arrangement lacks coherence.
Now Section 2 was rewritten and reorganized to present resistance mechanisms in a unified and logically structured progression. Content on biocides, permeability, and efflux systems has been merged and clarified to avoid abrupt transitions.
Specific Comments:
The following are some, but not all, of the suggestions/errors found in the article.
Comment 1. Only 14 serotypes are listed— please clarify or expand.
The text has been revised for accuracy. It’s current form: “Although E. coli comprises more than 180 O-serogroups, UPEC isolates are typically associated with a more limited core set of serotypes, most commonly O1, O2, O4, O6, O7, O8, O15, O16, O18, O21, O22, O25, O75 and O83. Of these, O25, O15 and O8 are the most encountered in UTIs.”
Comment 2. Please define the abbreviations in Fig. 1.
What does “ZUM” represent?
All abbreviations in Fig. 1 have been fully defined. The term “ZUM” has been changed with UTIs.
Comment 3. Please provide reference(s).
Appropriate references have been added.
Comment 4. “the only possible intervention” is too arbitrary. There are other non-pharmaceutical interventions, such as antimicrobial stewardship program and patient education.
We agree. The sentence war corrected such as: “In the context of current antibiotic crisis, bacteriophages are being explored as adjuncts to antibiotics for difficult-to-treat UPEC infections”.
Comment 5: (a) Correct to VB_ecoS-Golestan, not VB-ecoS-Golestan. (b) Please provide the reference for the statement and clarify why this particular phage therapy is highlighted over others.
The name has been corrected to VB_ecoS-Golestan, references were added, and the rationale for using this example—its detailed genomic characterization—was clearly stated.
Comment 6: Please spell out EPS upon first use.
Corrected. EPS is now spelled out as extracellular polymeric substances at first mention.
Comment 7: “as well as the resulting research for…….”: remove “resulting”
Revised accordingly.
Comment 8: The description regarding phage therapy gives the misleading impression that is is well-established with minimal concerns— please revise for accuracy.
The section has been rewritten to better reflect current limitations of phage therapy. “Phage therapy may benefit UTI treatment, including infections caused by MDR strains, but it is not yet fully established. Its limitations include narrow host specificity, variable activity among clinical isolates, and the risk of phage resistance [37].”
Comment 9: Please define the abbreviations in Fig. 2.Typo: VB_ecoS-Golestan
All abbreviations have been defined and the typo corrected. The description of the Figure has been considerably improved:
Figure 3. Pathogenesis, clinical challenges, and emerging therapeutic strategies against uropathogenic Escherichia coli (UPEC). UPEC initiates urinary tract colonization through fimbriae, adhesins, toxins, and siderophores, leading to infections such as cystitis and pyelonephritis. Biofilm formation enhances persistence and confers up to 1000-fold increased resistance, contributing to catheter-associated and chronic infections. Multidrug-resistant (MDR) UPEC strains exhibit resistance to β-lactams, aminoglycosides, and fluoroquinolones, resulting in higher treatment costs, longer hospitalizations, and frequent therapeutic failures. Novel therapeutic approaches include phage therapy using monophasic or cocktail formulations and phage-derived enzymes that degrade biofilms, and natural compounds such as phytotherapeutics, algae, and nanoparticles that inhibit quorum sensing, block adhesion, and promote biofilm disruption. Combination strategies integrating phages, antibiotics, and natural agents may reduce resistance development, shorten treatment duration, and prevent the spread of resistance genes. Abbreviations: UPEC – uropathogenic Escherichia coli; CAUTI – catheter-associated urinary tract infection; EPS – extracellular polymeric substances; QS – quorum sensing; MDR – multidrug-resistant; EGCG – epigallocatechin gallate.
Comment 10:Please spell out EMBO upon first use.
The manuscript after the improvements suggested also by other reviewers doesn’t contain a shortcut EMBO.
Comment 11: Typo
All noted typographical errors have been corrected.
Comment 12:
(a) Clarify: Enterobacterales order or Enterobacteriaceae family?
The sentence was correced as follows: “Genes encoding efflux pump proteins may be located on the bacterial chromosome, making this an intrinsic trait, or on plasmids, where they can be acquired through genetic material transfer during conjugation, transduction, or transformation”.
(b) Provide a more structured of efflux pump classifications and their relationship to MDR/XDR/PDR — consider merging this into the efflux section.
This section was reorganized, and efflux systems are now presented systematically.
(c) This sentence does not fit the context.
The sentence was removed to other place where it fits better, in a second paagraph of Section 2.
(d) This statement is overly simplified.
The statement was expanded and rewritten to ensure scientific accuracy.
“Resistance to one antimicrobial agent can, in some cases, extend to related drugs through mechanisms such as target modification, efflux pump overexpression or reduced membrane permeability. These forms of cross-resistance may broaden resistance profiles and decrease susceptibility to multiple classes of antimicrobial compounds (chemotherapeutics or disinfectants).”
Comment 13: If the section addresses chromosomal mutations affecting efflux pumps and porins that contribute to cross-resistance, clarify that these are not common resistance mechanisms in UPEC. Please introduce the resistance mechanisms more systemically.
We clarified that chromosomal mutations affecting porins and efflux pumps occur in UPEC but are not the predominant resistance mechanism.
Comment 14: Amoxicillin is neither the only antibiotics promoting biofilm formation nor specifically highlighted in the cited reference.
Corrected. The statement has been rewritten to avoid implying exclusivity.
Comment 15: The empirical treatment section appears to list antibiotics in a somewhat random manner. If ST131 is to be mentioned, it should be properly contextualized—highlighting its predominant role in the global dissemination of E. coli, as well as its hallmark traits of co-expressing key virulence determinants and harboring multidrug-resistant ESBL genes.
The section was reorganized, and ST131 is now properly contextualized, emphasizing its global spread and ESBL carriage.
Comment 16: The epidemiological reports cited are neither up to date nor reflective of the current clinical landscape.
In the revised manuscript, we expanded this section by incorporating both earlier reference studies-used solely as historical benchmarks-and more recent epidemiological evidence from diverse geographical regions. Specifically, we now contextualize previous findings from Poland (2013–2015) and Denmark (2014–2015) as baseline data and contrast them with more contemporary reports demonstrating substantially higher levels of resistance.
Comment 17: MDR and CR E. coli are typically discussed as distinct entities in epidemiological literature. Why is B2 phylogroup mentioned? — what is the specific relevance of these discussions to the subheading “bacterial strategies” ?
We revised this section to clarify the relevance of phylogroup B2 in the context of UPEC virulence and resistance and aligned it more precisely with the subheading.
Comment 18: This paragraph lacks relevance to resistance — consider merging it into the virulence section.
The paragraph has been moved to the virulence section.
Comment 19: Please clarify—do you mean: The response regulator (RR), which receives a phosphate group transferred from the SHK’s histidine residue to an aspartate residue “on the RR”?
This part was corrected to: “the response regulator (RR), which receives the phosphate group transferred from the SHK’s phosphorylated histidine to a conserved aspartate residue within the RR. Phosphorylated RR typically acts as a transcription factor, modulating the expression of target genes [79].
Comment 20: For this section designated as “The role of two-component regulatory systems in UPEC antibiotic resistance”—this is one of the few relevant descriptions that correlates to “TCS and resistance”. Please construct a more relevant and organized discussion.
This section has been reorganized to clearly present how TCS influence antibiotic resistance.
Comment 21: Please provide the reference.
Reference added.
Comment 22: The writing in this section needs refinement to improve readability.
The section has been thoroughly revised to improve clarity and readability.
Comment 23: Consider moving the paragraph to the plant-based approaches section.
Moved as suggested.
Comment 24: Describe the most common and well-described at the beginning of the section.
The entire section has been revised.
Comment 25: With which plant extracts?
Additional details and specific examples have been provided. “Dusane et al. (2014) showed that sub-inhibitory concentrations of the alkaloids piperine (Piper nigrum) and reserpine (Rauwolfia serpentina) decreased expression of fliC, motA and motB, reduced swimming and swarming motilities, and increased expression of adhesin genes (fimA, papA, uvrY) [117].”
Comment 26: This section is largely repetitive, please revise.
Repetitive elements were removed and the section has been streamlined.
Comment 27: This also repeats earlier content.
Revised to avoid redundancy and ensure clear differentiation between subsections.
Reviewer 3 Report
Comments and Suggestions for Authors
This is a timely, comprehensive, and well-researched review that effectively synthesizes a vast amount of information on a critically important topic. The manuscript successfully bridges the molecular mechanisms of UPEC antibiotic resistance with the promising potential of phytochemicals and alternative therapies. The scope is ambitious, covering resistance mechanisms, virulence, biofilms, efflux pumps, and a detailed survey of plant-based solutions. The literature cited is largely up-to-date, including references from 2024 and 2025. After revisions suggested in the attached PDF are to improve the narrative flow, balance the content, and strengthen the synthesis, this will be a valuable contribution to the field.

The manuscript requires thorough proofreading for minor grammatical errors and awkward phrasing.
Author Response
Response to Reviewer 3
We sincerely thank the Reviewer for the positive evaluation of our manuscript and for the constructive and insightful comments provided. Each suggestion has been carefully considered, and the revisions made in response have substantially improved the clarity, structure, and overall quality of the review. Below we address each point in detail.
For it's control
Thank you for the suggestion. The title has been updated to:“Current insights into antibiotic resistance in uropathogenic Escherichia coli and interventions using selected bioactive phytochemicals.”
The abstract is comprehensive but could be slightly more focused on the phytochemical theme promised by the title.
We appreciate this comment. The abstract has been revised to place stronger emphasis on phytochemical strategies, including their anti-adhesive, anti-motility, antibiofilm, quorum-sensing–inhibitory, and efflux-modulating effects. These modifications ensure that the abstract aligns more closely with the thematic focus of the review.
“Beyond conventional antibiotics, special emphasis is placed on phytochemical strategies as promising alternatives. Flavonoids, alkaloids, terpenoids, and essential oils exhibit antibacterial, anti-adhesive, and antibiofilm properties. These compounds modulate motility, suppress fimbrial expression, inhibit quorum sensing, and enhance antibiotic efficacy, acting both as independent agents and as adjuvants. Current in vitroand in vivo studies highlight the potential of plant-derived compounds and biologically based therapies to combat UPEC.”
Consider adding "biofilm," "efflux pumps," "alternative therapies" to improve discoverability.
Thank you for this recommendation. The keywords were revised to include “biofilm,” “efflux pump,” and “complementary therapies,” improving searchability and alignment with the manuscript’s central themes.
The introduction spends a significant amount of space on phage therapy (pages 3-5), which, while relevant, distracts from the paper's core focus on antibiotic resistance and phytochemicals. This material could be condensed and integrated into a dedicated subsection on alternative therapies later in the paper to maintain a sharper focus from the outset.
We agree with the Reviewer’s observation. The phage therapy section was condensed and reorganized.
The following sentence should be accompanied by a relevant reference.
The text has been corrected and appropriate references were added.
Please cite the studies too.
Thank you for noting this. The relevant citations have been added.
Please add relevant references here.
The section has been revised and the requested references have been incorporated - [8].
Here is needed to cite relevant studies.
We appreciate this comment. All relevant references have now been added - [9-10].
Following sentences should be accompanies by references.
Thank you for highlighting this. The text was corrected and supporting references were added to ensure accuracy and proper documentation.
“In the context of current antibiotic crisis, bacteriophages are being explored as adjuncts to antibiotics for difficult-to-treat UPEC infections. A well-chracterized example is the lytic phage VB-EcoS-Golestan, isolated from wastewater. Genomic analysis showed no lysogeny-associated genes or antibiotic-resistance genes, supporting its therapeutical potential, although clinical validation is still needed [19].”
The section "Current definitions of resistance, tolerance, and persistence" is well-written and useful. However, its connection to the rest of the review is not explicitly made. The authors should better integrate these concepts throughout the manuscript. For example, when discussing biofilms, explicitly state that they foster tolerance and persistence. When discussing phytochemicals that disrupt biofilms, frame them as potential tools to overcome tolerance.
We appreciate this insightful remark.
Section 2 “Current definitions of resistance, tolerance, and the persistence phenotype of bacteria to antimicrobial agents” has been removed and its content integrated into other relevant chapters to ensure a more streamlined structure – pages 4 and 8.
Page 4: “Antibiotics, as defined by the European Food Safety Authority (EFSA, 2025), selectively inhibit or kill microorganisms at low in vivo concentrations, distinguishing them from disinfectants. However, the misuse and overuse of antibiotics in medicine, agriculture, and veterinary practice have accelerated the emergence of resistant bacterial strains [17]. According to the European Committee on Antimicrobial Susceptibility Testing (EUCAST, 2024), bacterial isolates are classified as Susceptible (S), Susceptible, Increased Exposure (I), or Resistant (R) that refers to isolates that are likely to result in therapeutic failure even when exposure to the drug is increased. In fact, multidrug-resistant (MDR) bacteria is a major global public health challenge that has been noted, among other international organizations, by the World Health Organization (WHO), that has recently published an update of the Bacterial Priority Pathogens List, including third-generation cephalosporin- and carbapenem- resistant Enterobacterales in the most critical MDR group [18].”
Page 8: “Some researchers use the term recalcitrance to describe both tolerance and persistence, as these involve transient, non-heritable phenotypic adaptations that enable bacterial survival during antibiotic treatment. Recalcitrant cells can survive exposure to multiple antibiotic classes and contribute to the later emergence of highly resistant populations [43]. A key mechanism underlying both tolerance and persistence is dormancy, a physiological state in which bacterial cells reduce or halt metabolic activity. Because many antibiotics target active cellular processes, dormant cells are less susceptible. In tolerance, dormancy may be induced across the entire population, while in persistence, it is limited to a small subpopulation. Dormancy is associated with decreased energy production (ATP), and suppression of replication, transcription, and translation-contributing to the overall survival of the bacteria under antimicrobial pressure [44]. Understanding the distinctions between resistance, tolerance, and persistence is crucial for developing new therapeutic strategies, improving diagnostic accuracy, and mitigating the rise of treatment-refractory infections.”
This section is a valuable compilation of studies but reads largely as a list of "Compound X did Y in study Z." It lacks a powerful synthetic framework. Reorganize this section by the mechanistic actions of phytochemicals (e.g., "Anti-adhesive Phytochemicals," "Inhibitors of Motility and Biofilm," "Quorum Sensing Interference," "Efflux Pump Inhibition," "Membrane Disruptors") rather than just by plant source. Create tables to summarize this information, which would be immensely helpful for readers. For instance, a table listing key phytochemical classes, their sources, proposed mechanisms, and supporting evidence
Thank you for this valuable suggestion. The phytochemical section has been completely reorganized into mechanistic subsections:
5.1. Anti-adhesive phytochemicals
5.2. Phytochemicals as inhibitors of motility and biofilm formation
Additionally, we added Figure 4 with a detailed explanatory caption illustrating major mechanisms and Table 1, which summarizes representative phytochemicals, their sources, mechanisms of action, and supporting evidence. This restructuring greatly improves readability and conceptual clarity.
Figure 4. Phytochemical compounds targeting uropathogenic E. coli (UPEC) through multiple mechanisms. (A) Efflux pump inhibition: Selected phytochemicals and synthetic compounds such as reserpine, piperine, and harmaline act by inhibiting major efflux pump families (MF, RND), enhancing intracellular antibiotic accumulation and restoring susceptibility. (B) Anti-adhesive phytochemicals: Cranberry phenolics, resveratrol, and other plant-derived compounds interfere with UPEC adhesion and invasion, primarily through inhibition of FimH-mediated binding to uroepithelial cells. (C) Motility and biofilm inhibition: natural compounds including cinnamaldehyde, p-coumaric acid, carvacrol, and extracts from Betula pendula, Urtica dioica, Hedera helix var. glabra, Vaccinium vitis-idaea, and Cydonia oblonga suppress flagellar motility and biofilm formation via transcriptional modulation of motility and adhesion genes. (D) Membrane disruptors and essential oils: volatile oils from Cymbopogon citratus, Pelargonium spp., Eucalyptus spp., Calendula spp., and Melaleuca alternifolia exhibit bactericidal effects by disrupting bacterial membranes, with MBCs comparable to conventional antibiotics.
Table 1. Examples of plant-derived phytochemicals with anti-adhesive, anti-motility, and anti-biofilm activities against UPEC. The listed compounds interfere with bacterial virulence factors through enhancing antibiotic efficacy or preventing biofilm-associated resistance.
The summary is too brief given the length and depth of the review. It should be expanded to succinctly recap the main points: the severity of MDR UPEC, the key resistance mechanisms, and the most promising phytochemical and alternative strategies organized by their mechanism of action. It should also more forcefully outline the future challenges and directions (e.g., standardization of plant extracts, in vivo validation, clinical trials, bioavailability issues).
We thank the Reviewer for this helpful recommendation. The Summary section has been expanded to more clearly restate the global relevance of MDR UPEC and to synthesize the key resistance mechanisms discussed in the review. It now highlights the most promising phytochemical strategies according to their mechanisms of action and outlines major translational challenges, including issues of standardization, bioavailability, toxicity, and the need for in vivo validation and clinical trials. Additionally, we added a concise sentence emphasizing the potential synergy between phytochemicals and conventional antibiotics to enhance therapeutic relevance.
Round 2
Reviewer 1 Report
Comments and Suggestions for Authors
Authors have successfully addressed all the comments, so I am happy to recommend the publication of this manuscript after minor editorial modifications.
Reviewer 2 Report
Comments and Suggestions for Authors
Now the manuscript is acceptable.